# Insecticidal Toxicities of Three Main Constituents Derived from *Trachyspermum ammi* (L.) Sprague ex Turrill Fruits against the Small Hive Beetles, *Aethina tumida* Murray

**DOI:** 10.3390/molecules25051100

**Published:** 2020-03-01

**Authors:** Daniel Bisrat, Chuleui Jung

**Affiliations:** 1Agriculture Science and Technology Research Institute, Andong National University, Andong GB 36729, Korea; danielbisrat@gmail.com; 2Department of Pharmaceutical Chemistry and Pharmacognosy, School of Pharmacy, College of Health Sciences, Addis Ababa University, P.O. Box 1176 Addis Ababa, Ethiopia; 3Department of Plant Medicals, Andong National University, Andong GB 36729, Korea

**Keywords:** *Trachyspermum ammi* (L.) Sprague ex Turrill, contact toxicity, fumigant toxicity, small hive beetles, *Aethina tumida*, *p*-cymene, γ-terpinene, thymol

## Abstract

Small hive beetle (*Aethina tumida* Murray), indigenous to Africa, has spread to other parts of the world where has become a threat to the honeybee industry. In the present study, insecticidal properties (contact, fumigant, and repellent toxicities) of three main constituents derived from *Trachyspermum ammi* (L.) Sprague ex Turrill fruits essential oil were evaluated against adult small hive beetles under laboratory conditions. The Hydrodistillation of *T. ammi* fruits, grown in Ethiopia, yielded a pale yellow essential oil (3.5% *v*/*w*) with a strong aromatic odor. Analyses by gas chromatography-mass spectrometry identified twenty-two compounds that accounted for 98.68% of the total essential oil. The essential oil was dominated by monoterpenoids, comprising γ-terpinene (32.72%), *p*-cymene (27.92%), and thymol (24.36%). The essential oil showed strong contact and fumigation toxicities against the small hive beetle adults, with a LD_50_ value of 66.64 µg/adult and a LC_50_ value of 89.03 mg/L air, respectively. Among the main constituents, thymol was the most toxic component found in both contact (LD_50_ = 41.79 µg/adult) and fumigation (LC_50_ = 52.66 mg/L air) toxicities. The other two components, γ-terpinene and *p*-cymene, were less effective in both contact and fumigant toxicities testing. The results showed that *T. ammi* essential oil and thymol could serve as potential alternatives to synthetic insecticides for the control of small hive beetle adults.

## 1. Introduction

Animals pollinate over 60% of the world’s crop species, with honeybees accounting for a larger share [1,2]. Pathogens and pests, including ectoparasitic mites (*Varroa destructor* Anderson and Trueman, *Tropilaelpas mercedesae* Anderson and Morgan), and small hive beetles (SHB; *Aethina tumida* Murray), have been linked to long-term declines in bee populations [3].

The small hive beetle, *A. tumida* (Coleoptera: Nitidulidae), indigenous to Africa, has spread to other parts of the world where it has become a serious pest of honeybee colonies [4]. Adults and larvae of the small hive beetle primarily feed on brood, honey, and pollen, but can also cause fermentation and spoiling of the honey [5]. It is now understood that small hive beetles utilize volatiles emanating from the fermented pollen or honey, from adult worker bees (*Apis mellifera* Linnaeus), or other hive materials that allow them to locate honeybee colonies [6,7]. 

The small hive beetle was first detected in Miryang-si South Korea, in 2016 [8]. However, SHB’s invasion pathway to Korea was not clear until 2019. Recently, the origin of the SHB population in Korean has been linked to southeast USA [9]. Two control measures are commonly used for SHB in the United States, i.e., coumaphos strips that are placed inside the hive [10,11,12] and an insecticide soil drench that is applied to the soil surrounding the outside of the hive [12,13]. However, the use of these types of non-selective synthetic pesticides have been shown to have adverse effects on living organisms and the environment [14,15]. Thus, alternatives to traditional chemical treatments, such as botanical pesticides including essential oils that have lower mammalian and environmental toxicity, are being evaluated. 

Essential oils are well documented to show a broad spectrum of activity against pest insects, ranging from insecticidal, antifeedant, repellent, oviposition deterrent, growth regulatory, and antivector activities [16]. One plant essential oil that falls under this category is from *Trachyspermum ammi* (Bishop’s weed). The essential oil of *T. ammi* was reported to be one of the most effective plant extracts commonly used for control against *Zabrotes subfasciatus* (Mexican bean weevil) [17], *Sitophilus zeamais* (maize weevil) [18] and *Bactrocera oleae* (olive fruit fly) [19]. Indeed, these insecticidal activities of the essential oil were mainly attributed to the presence of thymol. Thymol is also a major constituent of some essential oils from *Thymus vulgaris* [20], *Carum copticum* [21], *Satureja* spp. [22], and other medicinal plants. In fact, thymol is known to possess various biological activities including antibacterial and antifungal activities [23], acaricidal properties [24], nematicidal activity [25], and insecticidal [26,27].

To our knowledge, few studies on insecticidal activity of plant extracts against the small hive beetles (SHBs) have been reported. In addition, the essential oil of *T. ammi* fruits and its main constituent have not been evaluated against small hive beetles. In the present investigation, we characterized the chemical composition of essential oil extracts from *T. ammi* fruit, and evaluated the insecticidal properties (contact, fumigant, and repellent toxicities) of its three main constituents against the small hive beetle, *Aethina tumida*.

## 2. Results

### 2.1. Characterization of the Essential Oil

Hydrodistillation of the fruits of *T. ammi* grown in Ethiopia gave a pale yellow essential oil, which was characterized by a strong aromatic odor with percentage yield of 3.5% (*v*/*w*, on a dry basis). The essential oil yield harvested in Ethiopia was consistent with the oil yield (2.0% to 4.4%) reported elsewhere [28]. The GC–MS analysis of the essential oil resulted in the identification of twenty-two compounds, representing 98.68% of the total essential oils (Table 1 and Appendix A). The essential oil was characterized by dominant levels of monoterpenoids, including γ-terpinene (32.72%), *p*-cymene (27.92%), and thymol (24.36%) (Appendix A). Therefore, the present study revealed that *T. ammi* essential oil from Ethiopia was characterized by a high level of γ-terpinene (γ-terpinene chemotype). However, the Indian *T. ammi* essential oil was dominated by relative high amounts of thymol (39.1%) (thymol-chemotype), followed by *p*-cymene (30.8%), and γ-terpinene (23.2%) [29].

### 2.2. Olfactory Orientation Responses of Adult SHBs 

Olfactory orientation assays of SHBs were conducted on odors of test samples in a dual-choice test using Y-tube, with a design similar to that reported by Li et al. [32]. Under red light, when SHBs were presented as a choice between *T. ammi* essential oil (treated arm) and acetone (control), they walked quickly back and forth between the arms, until the majority of SHB settled down at the far end of the control arm of the Y-tube. Individual SHB made a choice within the first minute. There was a significant difference in response of the repulsion of SHB when they were exposed to a choice between the odor of *T. ammi* essential oil and the control (*X*^2^ = 13.863, df = 3, and *p* = 0.003). Indeed, *T. ammi* essential oil had a repulsion rate of 74% ± 2.45% (Figure 1), with a response rate of 100% (zero no choices).

Following the repulsive response of *T. ammi* essential oil, the three main constituents were tested for their response toward SHBs. Thymol, one of the main components of *T. ammi* essential oil, showed a strong repulsion response (78% ± 3.74%) toward SHBs as compared with the control (*X*^2^ = 13.863, df = 5, and *p* = 0.017). It was also interesting to note that there was no significant difference in the repulsion rate of SHBs when they were presented with the odor of *T. ammi* essential oil and thymol (*X*^2^ = 3.314, df = 2, and *p* = 0.191). However, arithmetically, SHB’s repulsion rate to thymol tended to be higher than the *T. ammi* essential oil.

No significant repulsion response of SHBs was observed when SHB were presented a choice between γ-terpinene and the control in a dual Y-tube olfactory assay (*X*^2^ = 3.452, df = 4, and *p* = 0.485) (Figure 1). A similar pattern was observed when SHBs were exposed to a choice between *p*-cymene and the control (*X*^2^ = 0.680, df = 2, and *p* = 0.712) (Figure 1). These results indicated that *T. ammi* essential oil was a repellent to the SHBs, only due to the presence of thymol in the essential oil.

### 2.3. Contact Toxicity of Essential Oil and Its Main Constituents

The results of contact mortality assays are presented in Table 2. The results showed that the contact toxicity of *T. ammi* essential oil and its major components, γ-terpinene, *p*-cymene, and thymol against SHB adults varied. The essential oil showed a dose-dependent contact mortality, achieving 72% at the highest tested dose (100 µg/adult) (Figure 2). Following the promising activity of the essential oil (LD_50_ = 66.64 µg/adult), the main constituents of the essential oil were investigated for their contact toxicity.

Among the main components, thymol proved to be the most effective in its contact toxicity activity, with a LD_50_ value of 41.79 µg/adult. As shown in Figure 2, thymol had a positive dose-contact mortality response to SHBs with concentration. However, the other two, γ-terpinene and *p*-cymene, were less effective and achieved only LD_50_ values of 424.02 and 1208.71 µg/adult, respectively.

### 2.4. Fumigant Toxicity of Essential Oil and Its Main Constituents

A similar profile of fumigant toxicity, such as contact toxicity, was observed when the essential oil of *T. ammi* was subjected to fumigant activity against SHBs, with a LC_50_ value of 89.03 mg/L air (Table 3 and Figure 3). It is noteworthy that thymol, among the main constituents, exhibited a strong fumigant activity against SHBs with a LC_50_ value of 52.66 mg/L air, followed by γ-terpinene (522.11 mg/L air). *p*-Cymene was found to be the least toxic compound to SHBs, with a LC_50_ value of 1027.69 mg/L air. Therefore, the fumigant activity of *T. ammi* essential oil (EO) has been attributed to the thymol compound present in the essential oil. 

The difference in fumigant toxicity can be partly attributed to the high vapor pressure and phenolic nature of thymol. It was clearly demonstrated that higher vapor pressure of monoterpenes than diterpenes or sesquiterpenes has a positive coefficient of fumigant toxicity [33]. It is also noted that compounds that have a phenolic hydroxy group such as thymol are generally toxic to insects [34,35].

## 3. Discussion

*Trachyspermum ammi* fruits are traditionally used, in Ethiopia, to control pests. The insecticidal toxicities of *T. ammi* essential oil appear to be linked to its chemical constituents. The toxicity studies which are described here provide strong evidence that the insecticidal property of *T. ammi* essential oil to SHBs is attributed to the presence of thymol, which accounts for approximately 24.36% of the total essential oil (contact mortality LD_50_ = 41.79 μg thymol/adult and fumigant toxicity LC_50_ = 52.66 mg thymol/L air). It is interesting to note that thymol has also been reported to be toxic to several pests, effective to *Musca domestica* (housefly) and *Spodoptera litura* (tobacco cutworm) (LD_50_ = 25.4–29.0 μg/insect) [36,37] and also to *Drosophila melanogaster* (common fruit fly) and northern house mosquito, *Culex pipiens* Linnaeus [38,39]. On a different note, in our experiments, none of the essential oil of *T. ammi* and thymol dosages (100 mg/L air) resulted in significantly higher numbers of dead wandering larvae of SHBs in fumigation toxicity as compared with their controls. Therefore, we have ruled out thymol as an agent to control SHB’s larvae. 

Dahlgren et al. [40] showed that twenty-four hours after exposure, honeybees (both queens and workers) were more tolerant of thymol than conventional pesticides, with LD_50_ values of 3240 μg/g (583.2 μg/queen bee, queens) and 524 μg/g (56.6 μg/worker bee, workers). While taking this data into consideration [40], honeybee queens and workers were found to be 14-fold and 1.4-fold more tolerant of thymol, respectively, than SHBs as compared with per adults. Moreover, several thymol-based formulations have been used to control varroa mite and well tolerated by the bees [41]. Kanga and Somorin [42] demonstrated that the small hive beetle (SHB) was selectively susceptible to several classes of insecticides when subjected to the glass-vial bioassays. Coumaphos (LC_50_ = 1.61 μg/vial), which is currently used for control of SHB populations, was found to be less toxic to adult SHBs than fenitrothion (LC_50_ = 0.53 μg/vial), chlorpyrifos (LC_50_ = 0.53 μg/vial), and methomyl (LC_50_ = 0.54 μg/vial) [42]. In a previous report [43], thymol was found to be very effective in control of mites (*Varroa jacobsoni*) at a concentration between 5 and 15 μg/L air. Therefore, thymol can be considered as an agent worthy of being used in control of both mites and adult SHBs.

Considering the fact that the percentage composition of thymol in *T. ammi* essential oil can be increased from 24.4% to about 85.0% by a two-step reaction via aromatization of γ-terpinene to form *p*-cymene, followed by hydroxylation of *p*-cymene to offer thymol [44], this would make *T. ammi* essential oil one of the best sources of thymol.

The contact and fumigant activities of thymol were stronger than that of non-phenolic aromatic (*p*-cymene) and non-aromatic hydrocarbon (γ-terpinene). It was suggested that thymol potentiates GABA (γ-aminobutyric acid) receptors [45], and blockage of the GABA-gated chloride channel reduces neuronal inhibition which leads to hyper-excitation of the central nervous system, convulsions, and eventual death [46].

The percentage composition of the main constituents of the essential oil from *T. ammi* grown in Ethiopia varied from those growing elsewhere. It was characterized by a high content of γ-terpinene of Ethiopian origin (Trachyspermum ammi ct. γ-terpinene), while it was dominated by thymol in India (*Trachyspermum ammi* ct. thymol) [29]. Plant species produce different chemotypes for various reasons. This variation can be attributed to their geographical variations, i.e., the elevation at which a plant is grown, considering the evidence that *T. ammi* is grown in Ethiopia at 1700 to 2200 m altitude, whereas, it grows on the hills of Southeast Asia, at 750 m altitude [47]. In fact, the present study demonstrated that thymol is still one of the major constituents of *T. ammi* essential oil grown in Ethiopia, which accounts for 24.36% of the total essential oil.

## 4. Materials and Methods 

### 4.1. Materials

#### 4.1.1. Plant Material

The fruits of *T. ammi* were purchased from the local market (Shiromeda Market) in Addis Ababa, Ethiopia, in February 2019. The identity of the fruits was authenticated by Mr. Anteneh B. Desta at the National Herbarium, Department of Biology, Addis Ababa University.

#### 4.1.2. Insects

Small hive beetles (*Aethina tumida*) were reared in acrylic cages (38 × 38 × 34 cm) in the insectaria at the Insect Ecology Laboratory, Andong National University (ANU). The colony was maintained at 25 ± 2 °C, 60% ± 10% RH, and 12 L:12 D photoperiod on pollen dough. On preliminary testing, no significant differences between male and female responses of SHB were found with test samples (*p*> 0.05). Thus, adult beetles (either sex) were used for bioassays.

#### 4.1.3. Chemicals

Thymol (98.5%, CAS-No. 89-83-8), γ-terpinene (97%, CAS-No. 99-85-4) and *p*-cymene (99%, CAS-No. 99-87-6) were purchased from Sigma-Aldrich Korea (Cheoin-gu, Yongin city, South Korea).

#### 4.1.4. GC-MS Analysis

Gas chromatography-mass spectrometry (GC-MS) analysis of the essential oil was performed on an Agilent 7890B Gas Chromatography system (Agilent Technologies, USA) coupled to an Agilent 5977A Mass Spectrometer Detector system (Agilent Technologies). The HP5-MS type capillary column (a non-polar column; 30 m x 0.25 mm id and 0.25 µm film thickness, Agilent Technologies) was employed to separate and analyze individual constituents. Then, 1 µl of diluted sample (1/100; *v/v*, essential oil in acetone) was injected into the split mode with a 1:20 split ratio. The gas chromatographic conditions were carrier gas helium (1.0 mL/min), initial oven temperature 40 °C for 3 min isothermal, 40 to 150 °C at a rate of 6 °C/min, and 150 to 320 °C at a rate of 10 °C/min, and then held for 3 min. The injector temperature was set to 270 °C. Mass spectra were scanned in the range 40 to 500 amu with EI mode (70 eV) in full scan mode. The percentage composition of essential oil was calculated using the peak normalization method. The essential oil constituents were identified by comparing their retention indices (RI), mass spectra with NIST (National Institute of Standards and Technology), Adams library spectra, Wiley 7 n.1 mass computer library, and in published literature [30,31].

### 4.2. Methods

#### 4.2.1. Essential Oil Extraction

Fruits of *T. ammi* (250 g) were air dried and ground into a fine powder and subjected to hydrodistillation for 3 h using a Clevenger-type apparatus according to the standard procedure described in the European Pharmacopoeia (1997) [48]. The essential oils were collected, dried over anhydrous Na_2_SO_4_ and kept in airtight containers in a refrigerator at −4 °C until further analysis (8.7 mL). Essential oil yields were expressed in % (*v*/*w*), based on the weight of the dried plant material. 

#### 4.2.2. Olfactory Orientation Assay of SHBs

A Y-tube olfactometer system, previously described by Li et al. [32], was used to test the orientation responses of SHB towards test samples (essential oil, γ-terpinene, *p*-cymene, and thymol, each 1.25 ppm in acetone, separately) and control (acetone). The Y-tube olfactometer (Appendix A) was comprised of a central tube and two lateral arms (each 8 cm long and 18 mm internal diameter). Each arm was connected to an odor chamber (cryogenic vial, 4.3 cm in height and 3.0 cm inner diameter, 20 mL) holding the test sample. Each odor chamber had inlets for the incoming air and outlets for odors to exit the Y-tube. A charcoal-filtered and humidified air stream was passed into each arm at a flow rate of 250 mL/min (DK-800 air pump) to allow experimental odors to move towards the decision-making area. A mesh screen was placed at each of the endpoints of the olfactometer to prevent SHBs from escaping or coming in direct contact with the test samples. Small hive beetles, of either sex, were starved for over 2 h, before being tested. The test samples were separately applied to pieces of cotton. A 10 µl aliquot of each test samples was dripped onto a piece of cotton that had been placed inside one arm, and another piece of cotton permeated with 10 µl of acetone was placed in the other arm of the Y-section as a control. To prevent any positional bias in the behavior of the SHB, the relative position of the tested stimulus and its corresponding control were alternated between replicates. A clean Y-tube was used for each test in order to avoid carryover of odors. The Y-tube was illuminated with red light by means of an incandescent light bulb to preclude their use of visual cues during the experiments. One SHB at a time was introduced into the Y-tube after the airflow had been initiated. It was assumed to have made a choice when the SHB walked more than 2/3 the length of the treated source or control arm and stayed there for approximately 1 min or when it frequently visited the arm. A ‘‘no choice’’ decision was recorded if the SHB had not moved after 5 min. Each treated or control cotton was used only once and, then, was replaced with a fresh cotton for the next individual, and each individual SHB was used only once in the experiment. A total of 50 SHB were used for individual test samples. Response rate (%) and repulsion rate (%) of test samples on SHB in the Y-tube olfactometer were computed using a formula reported by Li et al. [32].
(1)$$\mathrm{Response}\;\mathrm{rate}\;\left(\%\right)\;=\;\frac{\mathrm{Responding}\;\mathrm{SHB}}{\mathrm{All}\;\mathrm{SHB}\;\mathrm{tested}}\;\times 100$$
(2)$$\;\mathrm{Repulsion}\;\mathrm{rate}\;\left(\%\right)\;=\;\frac{\mathrm{Responding}\;\mathrm{SHB}-\mathrm{SHB}\;\mathrm{showing}\;\mathrm{attraction}}{\mathrm{Responding}\;\mathrm{SHB}}\;\times \;100\;$$

#### 4.2.3. Contact Toxicity Assay

Contact toxicity of the essential oil from *T. ammi* fruits and its three main constituents (γ-terpinene, *p*-cymene, and thymol) were evaluated against the small hive beetle adults by topical application method, as described by Park et al. [49] with a few modifications (Appendix A). Preliminary studies were carried out to determine the appropriate test range concentrations. After SHB adults were cold anesthetized, aliquots of 1 µl of the test samples (20, 40, 60, 80, and 100 µg/SHB) were applied topically to the ventral abdomen of adult beetle using a microsyringe with repeating dispenser (Hamilton, Reno, NV, USA). Acetone (1 µl) was used as a control. After topical application of test samples, adult SHB were placed in a plastic Petri dish (1.5 cm in height and 5 cm diameter) with a piece of cotton containing honey solutions (ca. 0.5 mL of 5% (*v/v*) honey solution) and covered with a lid which had mesh-hole (1.4 cm diameter), thereby preventing fumigant effect of the tested samples. A total of 300 SHBs were used for each concentration with 5 adult SHBs per treatment in 10 replications. After 24 h treatment, mortality was checked. The LD_50_ values were calculated by using Probit analysis. 

#### 4.2.4. Fumigant Toxicity Assay

Fumigant toxicity of the essential oil and its three main constituents were also evaluated according to Park et al. [49] with a few modifications. The test was carried out using a plastic cylinder (6.5 cm in height and 3.5 cm inner diameter, 65 mL) containing a cryogenic vial (4.3 cm in height and 3.0 cm inner diameter, 20 mL) with mesh-holes at the bottom, and fitted approximately 2 cm above the bottom of the plastic cylinder (Appendix A). All test samples solutions were prepared in acetone at a concentration Ranging from 20 to 100 mg/L air. A paper disk (8 mm, Advantech) was impregnated with 10 µL of test sample solution. Disks were left to dry for 10 min to allow for acetone evaporation prior to being placed in the bottom lid of the cylinder. As a negative control, acetone (10 µL) only was applied. The top and bottom lids were sealed using parafilm to prevent leaking. Five adults per treatment were placed in a sieve with pollen dough (ca. 11 g), thereby preventing their direct contact with the test plant oils and compounds. The insects were maintained at 25 ± 1 °C and 80% relative humidity. After 24 h treatment, the insects and the pollen dough were moved to a new plastic Petri dish (4 cm in height a nd 9.6 cm diameter) and covered with a lid which had mesh-hole (4 cm diameter) for 10 min. The adult beetles were considered dead if their appendages did not move after air blown using an aspirator (Bug-Vac, Rose Entomology, AZ, USA). All treatments were replicated 5 times.

#### 4.2.5. Statistical Analysis

For each of the test samples, regression lines, LD_50_ or LC_50_ values, *χ*^2^ values, and 95% confidence limits were calculated for Y-tube olfactometer responses using the Probit analysis in SPSS version 16. A two-way ANOVA was used to compare contact and fumigant toxicities of test samples exposure (each individually) and the control on SHB mortality. Pairwise comparisons were made between test sample concentrations and the control. 

## 5. Conclusions

From the present study, the insecticidal toxicities of *T. ammi* essential oil seem to be mainly attributed to the presence of thymol. Owing to the fact that *T. ammi* can be easily cultivated, has high essential oil yield, and the other two major compounds (γ-terpinene and *p*-cymene) can be chemically converted into thymol via aromatization and hydroxylation reactions, respectively, this would make *T. ammi* essential oil a suitable candidate for the commercial development of a plant-derived insecticide. 

In light of the above results, we conclude that *T. ammi* essential oil and one of its main components (thymol) could be developed as fumigants or contact poisoners against the small hive beetle adult, *A. tumida*. 

## Figures and Tables

**Figure 1 molecules-25-01100-f001:**
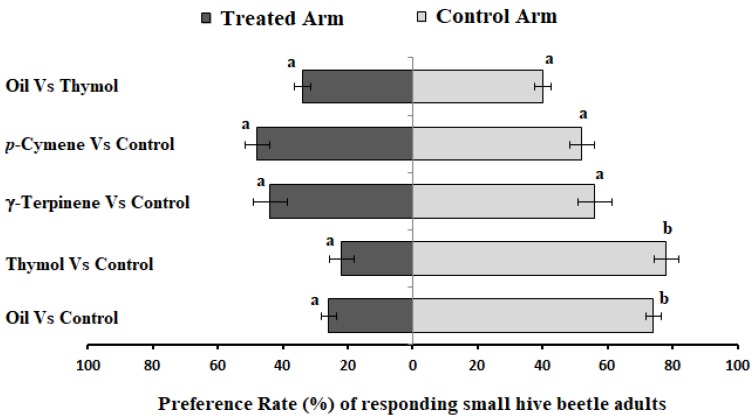
Response of the small hive beetle test in Y-tube olfactometer in dual choice b/n treatments and the control. N = 50 for all treatment. Note: The same letter in each test pair indicates no significance difference (*p* > 0.05), and different letters in the pair of test means there is a significance difference.

**Figure 2 molecules-25-01100-f002:**
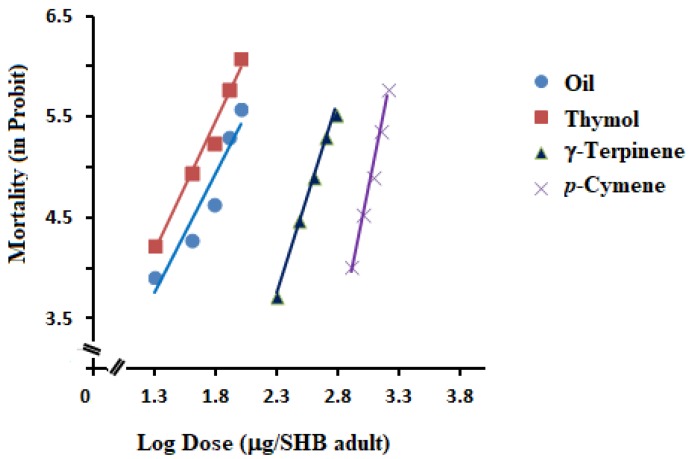
Dose-response lines of contact mortality of *Trachyspermum ammi* essential oil and its three main constituents (thymol, γ-terpinene, and *p*-cymene).

**Figure 3 molecules-25-01100-f003:**
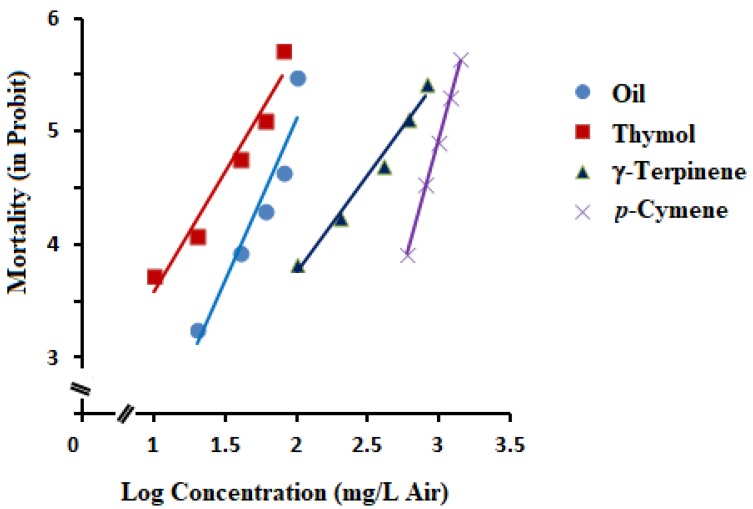
Concentration-response lines of fumigant mortality of *Trachyspermum ammi* essential oil and its three main constituents (thymol, γ-terpinene, and *p*-cymene).

**Table 1 molecules-25-01100-t001:** Chemical composition of the essential oil derived from the fruits of *T. ammi* grown in Ethiopia.

No.	Components ^a^	RI ^b^	RI ^c^	Percentage Composition	Methods of Identification
1	α-Thujene	926.3	927.8	0.79	MS, RI
2	α-Pinene	931.9	936.1	0.58	MS, RI
3	β-Pinene	974.8	977.7	4.56	MS, RI
4	β-Myrcene	991.8	989.2	1.07	MS, RI
5	3-Carene	1008.6	1011.3	0.12	MS, RI
6	α-Terpinene	1015.8	1017.1	0.55	MS, RI
7	*p*-Cymene	1028.1	1024.3	27.92	MS, RI
8	β-Phellandrene	1031.9	1030.0	0.94	MS, RI
9	γ-Terpinene	1062.7	1059.7	32.72	MS, RI
10	(*Z*)-Sabinene hydrate	1069.5	1066.5	0.14	MS, RI
11	(*E*)-Sabinene hydrate	1096.0	1098.1	0.17	MS, RI
12	Linalool	1099.6	1099.0	1.21	MS, RI
13	Undecane	1102.1	1100.0	0.12	MS, RI
14	(*Z*)-Verbenol	1149.5	1144.4	0.13	RI
15	*p*-Mentha-1,5-dien-8-ol	1166.0	1166.6	1.21	MS, RI
16	Terpinen-4-ol	1176.0	1177.1	0.85	MS, RI
17	α-Terpineol	1191.7	1189.7	0.28	MS, RI
18	(*Z*)-2,3-Epoxydecane	1264.2	-	0.15	MS
19	Thymol	1295.8	1290.1	24.36	MS, RI
20	Carvacrol	1302.3	1300.4	0.51	MS, RI
21	Thymol acetate	1358.7	1356.4	0.20	MS
22	Carvacrol acetate	1372.2	1373.1	0.10	MS
**Total**				**98.68%**	

Notes: ^a^ Compounds listed in order of elution. RI ^b^ are the Kovats retention indices were experimentally measured using homologous series of *n*-alkanes (C9–C29) on a non-polar (HP5-MS type column) capillary column under conditions listed in the Materials and Methods section; RI ^c^ are the Kovats retention indices taken from the literature [30,31]

**Table 2 molecules-25-01100-t002:** Lethal dose (LD_50_) and 95% confidence limits (CL) estimated for the small hive beetles 24 h after topical application of *T. ammi* essential oil and its three main constituents.

Test Samples	Probit Analysis						
	N	LD_50_ (µg/adult)	95% CL	Slope ± SE	Intercept	*X* ^2^	Df
Essential oil	300	66.64	57.12–80.21	2.50 ± 0.75	0.45	20.432	48
Thymol	300	41.79	34.74–48.62	2.58 ± 0.72	0.82	16.856	48
γ-Terpinene	300	424.02	382.72–476.13	3.77 ± 1.09	−4.91	19.250	48
*p*-Cymene	300	1208.71	1129.97–1297.72	5.80 ± 1.66	−12.86	23.644	48

**Table 3 molecules-25-01100-t003:** Lethal concentration (LC_50_) and 95% confidence limits (CL) estimated for the small hive beetles, 24 h after fumigation exposure to *T. ammi* essential oil and its three main constituents.

Probit Analysis							
Test Samples	N	LC_50_ (mg/L air)	95% CL	Slope ± SE	Intercept	*X* ^2^	Df
Essential oil	300	89.03	76.75–110.93	3.01 ± 0.94	−0.86	31.800	48
Thymol	300	52.66	43.62–66.78	2.14 ± 0.60	1.31	26.497	48
γ-Terpinene	300	522.11	418.55–700.35	1.77 ± 0.56	0.20	18.409	48
*p*-Cymene	300	1027.69	945.24–1126.55	4.61 ± 1.36	−8.89	19.859	48

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
