# Peer review of "Insecticidal Toxicities of Three Main Constituents Derived from Trachyspermum ammi (L.) Sprague ex Turrill Fruits against the Small Hive Beetles, Aethina tumida Murray"

_molecules, 2020, doi:10.3390/molecules25051100_

Round 1
Reviewer 1 Report
The authors present a thorough examination of the toxic effect of Trachyspermum ammi essential oil extracts to small hive beetles. Their results indicate that thymol is the main constituent with insecticidal activity and propose that it could be used to control small hive beetles in honey bee nests. The main issue I have with the results the way they are presented is that there is already a body of literature on the insecticidal properties of thymol. What the authors need to expand on here is a comparison of dose response to thymol by both SHBs and honey bees. They mention that honey bees are somewhat tolerant to thymol, but are they tolerant to the level needed to control SHBs?
Overall, the study methods are well described. Grammar and sentence structure still need quite a bit of work though. I have made a number of corrections in the specific areas below, but these are not comprehensive.
Specific comments and edits are as follows:
Abstract:
Lines 18 and 29: abbreviations are generally not used in the abstract.
Introduction:
Lines 36 and 38: add author name (Murray) after species name the first time it is written (i.e. line 36) and use A. tumida without author name for line 38.
Line 39: change became to has become.
Line 39: do not start a sentence with an abbreviation. Change sentence to – Small hive beetle was first detected in Miryang-si South Korea in 2016.
Line 42: delete was
Line 43: beetles should be beetle
Line 43: why is optimum survival temperature mentioned here? Expand on this or delete.
Line 44: change to, but can also cause fermentation and spoiling of the honey.
Line 45: give author name for Apis mellifera. (also check that all other species names have author listed the first time the species is mentioned).
Line 45: change to, beetles utilize volatiles emanating from the fermented pollen or honey, from adult worker bees (Apis mellifera), or other hive materials that allow…
The introduction section is very disjointed and needs some reorganization. For example, move section about distribution (lines 39-43) below life history section and use it to start the paragraph beginning at line 48.
Line 48: change to, Two control measures are commonly used for SHB in the United States, coumaphos strips that are placed inside the hive and insecticidal soil drench applied to the soil surrounding the outside of the hive.
Line 50: add, these types of before non-selective; change has to have been; organism should be organisms.
Line 53: change sought to evaluated.
Line 56: delete of such; add the word from before the plant species name
Line 58: change sentence to be, one of the most effective compounds commonly used for control against …
Line 59: The mention of Thymol here feels random. Put it in context or delete.
Line 61: spp should be spp. since it is an abbreviation
Line 64: change fewer to few.
Line 66: delete Therefore
Line 67: change to, characterized the chemical composition of essential oil extracts from T. ammi fruit, and …
Results:
Line 111: change to, The same letter in each test pair indicates…
Line 114: add a comma after oil
Line 125: change activity to toxicity
Line 129: delete, of
Line 133: delete at; abbreviate T. ammi
Line 136: delete, on a particular note and start the sentence with Among
Line 159: Why start abbreviating essential oil now? This just makes things confusing. Be consistent.
Line 168: abbreviate T. ammi
Materials and Methods:
Lines 256-267: ° C in this paragraph have the degree sign underlined for some odd reason?
Line 267: change to, in published literature
Line 271: is hydrodistillation one word or two? Make sure you are consistent throughout.
Line 274: delete, for
Line 278: add a comma after system and after [33]
Line 281: add was comprised
Lines 286-287: change to, prevent SHB from escaping or coming in direct contact with the test samples.
Line 287: move the part about differences between sexes to the insect section 4.1.2
Line 313: change testing to test
Lines 314-315: Begin sentence with, After SHB adults were cold anesthetized, aliquots of …
Line 330: change sentence to, Disks were left to dry for 10 minutes to allow for acetone evaporation prior to being placed in the bottom lid of the cylinder.
Line 332: delete, from; change per a treatment to per treatment
Line 335: what bean are you referring to?
Lines 198-203: This needs more discussion. What is the study you are referring to and why are you selecting it for comparison to your results here?
Lines 209-211: Start the discussion section with this part about honey bee tolerance to thymol. If you are proposing this as a potential control for SHB you need to give as much evidence as possible that the treatments are safe for the bees (or at least safer than other insecticides that are used).
Supplementary Figures:
S3 – the images are not very clear and it is difficult to see what is going on in (A) specifically.
Author Response
Dear Editor:
We have uploaded the revised version of our manuscript (coded Manuscript ID: molecules-725097) as per the comments of the reviewers. For the sake of your quick scrutiny, all changes that have been made are clearly highlighted red using the "Track Changes" function in Microsoft Word. Point-by-point replies to the queries raised by the three reviewers are listed below.
Point-by-point responses
Response to Reviewer-1 Report
Comments and Suggestions for Authors
- The authors present a thorough examination of the toxic effect of Trachyspermum ammi essential oil extracts to small hive beetles. Their results indicate that thymol is the main constituent with insecticidal activity and propose that it could be used to control small hive beetles in honey bee nests. The main issue I have with the results the way they are presented is that there is already a body of literature on the insecticidal properties of thymol. What the authors need to expand on here is a comparison of dose response to thymol by both SHBs and honey bees. They mention that honey bees are somewhat tolerant to thymol, but are they tolerant to the level needed to control SHBs?
- Revised to include some of the comments raised by the reviewer (Refer to lines 203-232)
- Overall, the study methods are well described. Grammar and sentence structure still need quite a bit of work though. I have made a number of corrections in the specific areas below, but these are not comprehensive.
- Corrections done as per suggestion
- Lines 18 and 29: abbreviations are generally not used in the abstract.
- All abbreviations are removed from the abstract section (Refer to Lines 15-30).
- Lines 36 and 38: add author name (Murray) after species name the first time it is written (i.e. line 36) and use tumidawithout author name for line 38.
- Revised accordingly (Refer to lines 36-39)
- Line 39: change became to has become.
- Corrected (Refer to line 40)
- Line 39: do not start a sentence with an abbreviation. Change sentence to – Small hive beetle was first detected in Miryang-si South Korea in 2016.
- Corrected (Refer to line 50)
- Line 42: delete was
- Done (Refer to line 52)
- Line 43: beetles should be beetle
- Corrected (Refer to line 44)
- Line 43: why is optimum survival temperature mentioned here? Expand on this or delete.
- Sentence about “optimum survival temperature…” has been now removed. (Refer to lines 43-44)
- Line 44: change to, but can also cause fermentation and spoiling of the honey.
- Corrected (Refer to line 45)
- Line 45: give author name for Apis mellifera. (also check that all other species names have author listed the first time the species is mentioned).
- Done (Refer to lines 48; 36-37)
- Line 45: change to, beetles utilize volatiles emanating from the fermented pollen or honey, from adult worker bees (Apis mellifera), or other hive materials that allow…
- Corrected (Refer to lines 46-48)
- The introduction section is very disjointed and needs some reorganization. For example, move section about distribution (lines 39-43) below life history section and use it to start the paragraph beginning at line 48.
- Reorganized (Refer to lines 50-52)
- Line 48: change to, Two control measures are commonly used for SHB in the United States, coumaphos strips that are placed inside the hive and insecticidal soil drench applied to the soil surrounding the outside of the hive.
- Revised (Refer to lines 54-56)
- Line 50: add, these types of before non-selective; change has to have been; organism should be organisms.
- Done (Refer to lines 56-57)
- Line 53: change sought to evaluated.
- Changed (Refer to line 60)
- Line 56: delete of such; add the word from before the plant species name
- Done (Refer to line 63)
- Line 58: change sentence to be, one of the most effective compounds commonly used for control against …
- Done (Refer to line 65)
- Line 59: The mention of Thymol here feels random. Put it in context or delete.
- Reorganized (Refer to lines 67-58)
- Line 61: spp should be spp. since it is an abbreviation
- Done (Refer to line 69)
- Line 64: change fewer to few.
- Corrected (Refer to line 72)
- Line 66: delete Therefore
- Done (Refer to line 74)
- Line 67: change to, characterized the chemical composition of essential oil extracts from ammifruit, and …
- Done (Refer to line 75)
- Line 111: change to, The same letter in each test pair indicates…
- Done (Refer to line 119)
- Line 114: add a comma after oil
- A comma is added (Refer to line 122)
- Line 125: change activity to toxicity
- Changed (Refer to line 133)
- Line 129: delete, of
- Done (Refer to line 137)
- Line 133: delete at; abbreviate ammi
- Done (Refer to line 141)
- Line 136: delete, on a particular note and start the sentence with Among
- Done (Refer to line 144)
- Line 159: Why start abbreviating essential oil now? This just makes things confusing. Be consistent.
- Corrected (Refer to line 167)
- Line 168: abbreviate ammi
- Done (Refer to line 176)
- Lines 256-267: ° C in this paragraph have the degree sign underlined for some odd reason?
- Corrected (Refer to lines 267-278)
- Line 267: change to, in published literature
- Corrected (Refer to line 278)
- Line 271: is hydrodistillation one word or two? Make sure you are consistent throughout.
- It’s one word, and corrected accordingly (Refer to lines 282-283)
- Line 274: delete, for
- Corrected (Refer to line 285)
- Line 278: add a comma after system and after [33]
- A comma is added (Refer to line 289)
- Line 281: add was comprised
- Corrected (Refer to line 291)
- Lines 286-287: change to, prevent SHB from escaping or coming in direct contact with the test samples.
- Corrected (Refer to line 298)
- Line 287: move the part about differences between sexes to the insect section 4.1.2
- Moved to section 4.1.2 (Refer to lines 258-260)
- Line 313: change testing to test
- Corrected (Refer to line 323)
- Lines 314-315: Begin sentence with, After SHB adults were cold anesthetized, aliquots of …
- Corrected (Refer to lines 323-324)
- Line 330: change sentence to, Disks were left to dry for 10 minutes to allow for acetone evaporation prior to being placed in the bottom lid of the cylinder.
- Corrected (Refer to line 342-343)
- Line 332: delete, from; change per a treatment to per treatment
- Corrected (Refer to line 344)
- Line 335: what bean are you referring to?
- Corrected (Refer to line 347)
- Lines 198-203: This needs more discussion. What is the study you are referring to and why are you selecting it for comparison to your results here?
- Revised as per suggestion (Refer to lines 203-215)
- Lines 209-211: Start the discussion section with this part about honey bee tolerance to thymol. If you are proposing this as a potential control for SHB you need to give as much evidence as possible that the treatments are safe for the bees (or at least safer than other insecticides that are used).
- Revised (Refer to lines 220-232)
- Supplementary Figures:
- S3 – the images are not very clear and it is difficult to see what is going on in (A) specifically.
- (A) is now replaced by new figure (Refer to file-Mol-725097-supplem-Revised)
Reviewer 2 Report
The paper is good for the publication and interesting for the scientists anyway some revision are need.
4.1.3 The purity of the reference subtance should be given and the concentration of the stock solution should be given.
4.2 The performance of the method should be given for example the recoveries, standard deviation number of replicates, fortification levels.
4.1.4 The ions used to quantify and qualifier the substances should be given.
Author Response
Dear Editor:
We have uploaded the revised version of our manuscript (coded Manuscript ID: molecules-725097) as per the comments of the reviewers. For the sake of your quick scrutiny, all changes that have been made are clearly highlighted red using the "Track Changes" function in Microsoft Word. Point-by-point replies to the queries raised by the three reviewers are listed below.
Point-by-point responses
Response to Reviewer-2 Report
- The paper is good for the publication and interesting for the scientists anyway some revision are need.
- 1.3 The purity of the reference subtance should be given and the concentration of the stock solution should be given.
- The purity and CAS Number of all the reference are now given (Refer to lines 266-269)
- 2 The performance of the method should be given for example the recoveries, standard deviation number of replicates, fortification levels.
- We have used normalization method, not calibration method. The percentage composition of essential oil was calculated using the peak normalization method (Refer to lines 280-281)
- 1.4 The ions used to quantify and qualifier the substances should be given.
- Full scan ion mode is used. Not selective ion monitoring (SIM) used (Reefer to line 280)